# Has it really improved?
# Knowledge Graph Based Separation and Fusion for Recommendation

## Abstract

In this paper we study the knowledge graph (KG) based recommendation systems. We first design the metric to study the relationship between different SOTA models and find that the current recommendation systems based on knowledge graph have poor ability to retain collaborative filtering signals, and higher-order connectivity would introduce noises. In addition, we explore the collaborative filtering recommendation method using GNN and design the experiment to show that the information learned between GNN models stacked with different layers is different, which provides the explanation for the unstable performance of GNN stacking different layers from a new perspective. According to the above findings, we first design the model-agnostic Cross-Layer Fusion Mechanism without any parameters to improve the performance of GNN. Experimental results on three datasets for collaborative filtering show that Cross-Layer Fusion Mechanism is effective for improving GNN performance. Then we design three independent signal extractors to mine the data at three different perspectives and train them separately. Finally, we use the signal fusion mechanism to fuse different signals. Experimental results on three datasets that introduce KG show that our KGSF achieves significant improvements over current SOTA KG based recommendation methods and the results are interpretable.

## 1 Introduction

The recommendation system is the important technique for information filtering, which can help users find the data they want in a large amount of data. Collaborative filtering algorithm is a classical recommendation method, and its main idea is to make recommendations by mining the collaborative signals between users and items. As a deep learning method, graph neural networks (GNN) have been used to effec-

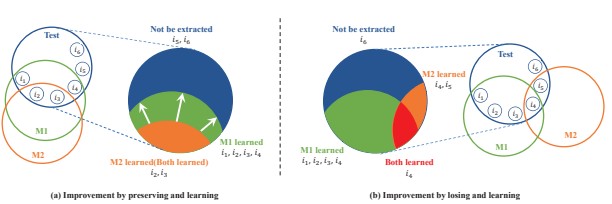

Figure 1: Two categories of improvement

tively mine users' collaborative signals, such as NGCF (Wang et al., 2019d), LightGCN (He et al., 2020). Recent works (Wu et al., 2021; Yu et al., 2022) use LightGCN as the backbone to introduce contrastive learning and achieve better performance.

From LightGCN (He et al., 2020) to SimGCL (Yu et al., 2022), the performance is constantly improving. For the convenience of description, we assume that there are two models, `M1` and `M2`. The overall performance of `M1` is better than that of `M2`. In practical applications, for some specific users, the recommendation effect of `M1` may be inferior to that of `M2`. A problem that may be overlooked is, is `M1` really an improvement over `M2` ? That is, what is the relationship between `M1` and `M2`? Figure 1 shows two possible relationships between `M1` and `M2`. ($i_1 \sim i_6$ are commodities, the circle "`Test`" represents the range of the test set, the circle "`M1`" represents the range of top K commodities given by the `M1` model, the cricle "`M2`" represents the range of the test the top K commodities given by the `M2` model). In Figure 1(a), `M1` learns new things on the basis of retaining the infor-

mation of `M2`, while in Figure 1(b) the improvement of `M1` to `M2` is to lose part of the information of `M2` and learn more new information. To objectively measure these two cases, we design a new metric, `Intersection@N`, to measure the differences between two models. Based on this indicator, we conduct experiments between different collaborative filtering models (Wu et al., 2021; Yu et al., 2022), and between different stacked layers of the same collaborative filtering model (which are described in Section 2.1 and Appendix C), and obtain two findings:(1) The relationship between different collaborative filtering methods using GNN is shown in Figure 1(b). (2) The relationship between models based on collaborative filtering that uses the same GNN model but stacks different layers is shown in Figure 1(b), i.e., models that stack higher layers cannot fully "include" models that stack lower layers.

Many studies (Wang et al., 2019d; He et al., 2020) pointed out that within a certain range, the more GNN layers are stacked, the higher the model performance is. The performance decreases after exceeding this range. Numerous studies (He et al., 2020; Zhao & Akoglu, 2019) attribute the poor performance to the over-smoothing of nodes caused by multi-layer stacking, and based on this reason, many methods are designed to alleviate the over-smoothing. A common feature of these works is that they all choose a model with fixed stacked L layers as the final model. The default assumption in doing so is that the model with good performance (stacked L layers) is an improvement over the model with stacked T layers of poor performance and the improvement is understood as in Figure 1(a). However our experiments show that this assumption is invalid and instead their relationship should be as shown in Figure 1(b) (i.e. The first finding). **The first observation of this paper is that the recommendation method based on collaborative filtering using GNN does not fully exploit the performance of GNN**.

Accompanying GNN, knowledge graphs (KG) are introduced into the recommendation systems to improve their performances with auxiliary information. The popular KG based recommendation methods are KGAT (Wang et al., 2019c) and KGIN (Wang et al., 2021), which connect KG and user-item bipartite graphs through items, thus unify the two into one graph structure.

According to KGAT and KGIN, we believe that the KG based recommendation system includes **item-based collaborative signals**, **content signals** and **attribute-based collaborative signals**. These three signals are mined in KGAT and KGIN. The first two signals are mined in the User-Item bipartite graph and the KG respectively, and the third signal is mined in the unified graph by higher-order connectivity (Wang et al., 2019c). However, we have two observations with this unified graph structure: **(1) Poor preservation for collaborative filtering signals**. We use the `Intersection@N` metric to measure the KG-based recommendation method and the collaborative filtering recommendation method, and find the relationship between the two methods are as shown in Figure 1(b). Existing methods that introduce KG discard half of the information based on collaborative filtering and learn more information introduced by KG, so the performance of the former is higher than that of the latter. **(2) Unnecessary information are introduced by higher-order connectivity which makes the propagation path too long**. Taking user $u_2$ recommending $i_4$ in Figure 5 as an example, a possible path of a path is $u_2 \xrightarrow{like} a_3 \xrightarrow{r_2} i_4$, and the semantic information of this path is that $u_2$ likes items with $a_3$ attribute. However, the path given by KGAT is $u_2 \xrightarrow{like} i_2 \xrightarrow{r_1} a_3 \xrightarrow{r_2} i_4$. In this path, the information of node $i_2$ includes content signal and user-based collaborative signal, which is not helpful for the original semantic information and will introduce unnecessary information. In addition, longer propagation paths also introduce noise.

Based on above three observations, in this paper we propose a general knowledge graph based separation and fusion model. It consists of three core parts to meet the three challenges mentioned above:

**Cross-Layer Fusion Mechanism.** We find that there are differences in the information learned between models that stack different layers, so we cannot directly select a model that stacks N layers. We design a model-agnostic, general-purpose Cross-Layer Fusion Mechanism without any trainable parameters, which fuses models stacked with different layers and can preserve the information of different models.

**Signal Extractor.** We design three independent, separately trained signal extractors to extract the three kinds of signals in the data mentioned before, which can avoid the mutual influence of each signal. We use the existing collaborative filtering method for item-based collaborative signal extraction and the Cross-Layer Fusion Mechanism is applied to further improve the performance. For

attribute-based collaborative signals, we process the original User-Item-Attribute graph into a User-Attribute-Item graph and apply LightGCN to extract signals from the User-Attribute graph. The Cross-Layer Fusion Mechanism is also applied. The extraction of the content signal draws on the idea of Transformer (Vaswani et al., 2017), which uses the user as the Query Vector and the attribute as the Key Vector to obtain the user's interest in each attribute to get a fine-grained explanation.

**Signal Fusion Mechanism**. The three signal extractors get different scores. We draw on the idea of ensemble learning and design Signal Fusion Mechanism, which is essentially weighted summation.

We conduct extensive experiments on three real datasets. Experimental results show that our Cross-Layer Fusion Mechanism improves collaborative filtering better than state-of-the-art methods, such as SGL Wu et al. (2021), SimGCL Yu et al. (2022). Our designed KGSF (**K**nowledge **G**raph based **S**eparate and **F**usion model) also outperforms state-of-the-art methods in knowledge graph-based recommendation, such as KGAT (Wang et al., 2019c), KGIN (Wang et al., 2021), KGCL (Yuhao et al., 2022), HAKG (Du et al., 2022). The **contributions** of this paper are summarized as follows: **(1)** The Intersection indicator is designed to measure the relationship between different models. Using this indicator we give a new explanation for the unstable performances of GNNs stacking different layers, and find that the improved relationship between most models belongs to Figure 1(a). **(2)** The Cross-Layer Fusion Mechanism can effectively improve the performance of GNN, which is model-independent, general, and without any training parameters. **(3)** A highly interpretable and extensible KGSF framework is proposed. **(4)** Experimental studies on three datasets demonstrate the superiority and effectiveness of Cross-Layer Fusion Mechanism and KGSF.

## 2 PRELIMINARIES

It triggered a research boom since NGCF (Wang et al., 2019d) introduced GNN into the collaborative filtering method. LightGCN (He et al., 2020) removes activation and projection in NGCF and achieves remarkable results. Afterwards, contrastive learning was introduced and the SGL (Wu et al., 2021) framework was proposed. Then there is the SimGCL (Yu et al., 2022) method, which removes graph augmentations and adds perturbation on the basis of contrastive learning. The performance of the above method gradually increases.

However, there are two problems to be studied. First, from NGCF and LightGCN to SGL and SimGCL, is their relationship shown in Figure 1(a) or Figure 1(b)? Second, the increase in the number of GNN layers will degrade the performance. Many studies attribute the reason to the learned embedding tends to be smooth. Is the relationship between these different layer shown in Figure 1(a) or Figure 1(b)? Here, we choose two SOTA methods SGL (Wu et al., 2021) and SimGCL (Yu et al., 2022) for comparative experiments, and give conclusions in Appendix C and 2.1, respectively. Meanwhile, a solution is given in subsection 2.2. In addition, we suggest readers to read concepts including "Intersection", "UA Graph", "IA Graph", etc. in Appendix B.

### 2.1 THE RELATIONSHIP BETWEEN DIFFERENT LAYERS IN THE SIMGCL AND SGL

A large number of GNN-based (Wang et al., 2019d; He et al., 2020) collaborative filtering methods believe that the performance will increase with the increase of the number of layers in a small range. However, the performance will decrease when the stacking reaches a certain level. Table 2 also confirms this. Many works (He et al., 2020; Zhao & Akoglu, 2019; Rong et al., 2019) attribute the poor performance to the fact that as the number of layer increases, the node representations learned by GNN become smoother and thus lack discrimination. The default premise of this view is that a model with better performance is an extension of a model with poor performance, in which the former perfectly preserves the latter's information. To verify this, we still use Intersection@20 metric to

Table 1: Intersection@20 between different layers.

|        |     | Yelp2018 |        |        | Last-FM |        |        | Amazon-Book |        |        |
|--------|-----|----------|--------|--------|---------|--------|--------|-------------|--------|--------|
|        |     | L-1      | L-2    | L-3    | L-1     | L-2    | L-3    | L-1         | L-2    | L-3    |
|        | L-1 | 1.0000   | 0.8284 | 0.7528 | 1.0000  | 0.7198 | 0.6398 | 1.0000      | 0.8101 | 0.7330 |
| SGL    | L-2 | 0.7791   | 1.0000 | 0.7838 | 0.7296  | 1.0000 | 0.6235 | 0.7027      | 1.0000 | 0.7371 |
|        | L-3 | 0.6873   | 0.7601 | 1.0000 | 0.6000  | 0.5785 | 1.0000 | 0.6172      | 0.7154 | 1.0000 |
|        | L-1 | 1.0000   | 0.7945 | 0.7758 | 1.0000  | 0.7296 | 0.6661 | 1.0000      | 0.8179 | 0.8190 |
| SimGCL | L-2 | 0.7583   | 1.0000 | 0.8494 | 0.7251  | 1.0000 | 0.7459 | 0.7922      | 1.0000 | 0.8751 |
|        | L-3 | 0.7381   | 0.8462 | 1.0000 | 0.7126  | 0.8040 | 1.0000 | 0.7657      | 0.8443 | 1.0000 |

conduct experiments between different layers of the same method, and the results are shown in Table 1. We summarize our observation and conclusions as follows: (1) We found that on the whole, the similarity between different layers of the same method and the same dataset is between 57% and 85%. This shows that the information learned by different layers is also partly different and the relationship between them is intertwined as shown in Figure 1(a). In other words, the layer with better performance is not an improvement on the layer with worse performance. One possible reason for the poor performance of the model with higher layers is that the total amount of high-level information in the dataset is smaller than low-level information. (2) We also found that the minimum of similarity between layers occurs between layers 1 and 3, except that the minimum of the SGL on the Last-FM dataset occurs between layers 2 and 3. This shows that with the increase of the number of layers, the retention ability of the upper layer to the lower layer is gradually reduced, and the new information learned by the upper layer is greater than the reduced. Therefore, the effect of the upper layer is better than that of the lower layer.

## 2.2 CROSS-LAYER FUSION MECHANISM

Heretofore, many methods (Zhao & Akoglu, 2019; Rong et al., 2019; Liu et al., 2020; Feng et al., 2020; Chen et al., 2020) have been proposed to mitigate the performance degradation caused by increasing the number of layers. However, this is based on the premise that the better-performing model retains the information of the poorer-performing model. Our experiments prove that this premise is wrong. According to the experiments in section 2.1, we found that the information learned between different layers of the same model is independent. Therefore, it's naive to think

Table 2: Performance comparison between SGL and SimGCL.

|  |  | Yelp2018 | | Last-FM | | Amazon-Book | |
|---|---|---|---|---|---|---|---|
|  |  | Recall | NDCG | Recall | NDCG | Recall | NDCG |
| SGL | Layer-1 | 0.0723 | 0.0477 | 0.0782 | 0.0700 | 0.1251 | 0.0702 |
|  | Layer-2 | 0.0773 | 0.0508 | 0.0834 | 0.0731 | 0.1431 | 0.0803 |
|  | Layer-3 | 0.0796 | 0.0520 | 0.0839 | 0.0718 | 0.1481 | 0.0826 |
|  | Fusion | **0.0799** | **0.0522** | **0.0931** | **0.0810** | **0.1599** | **0.0895** |
|  | Imp% | 0.38% | 0.38% | 10.97% | 10.81% | 7.97% | 8.35% |
| SimGCL | Layer-1 | 0.0789 | 0.0515 | 0.0690 | 0.0601 | 0.1445 | 0.0777 |
|  | Layer-2 | 0.0824 | 0.0541 | 0.0701 | 0.0611 | 0.1500 | 0.0816 |
|  | Layer-3 | 0.0828 | 0.0543 | 0.0699 | 0.0601 | 0.1550 | 0.0845 |
|  | Fusion | **0.0839** | **0.0550** | **0.0763** | **0.0669** | **0.1603** | **0.0869** |
|  | Imp% | 1.33% | 1.29% | 8.84% | 9.49% | 3.42% | 2.72% |

that simply retaining the information between different layers improves performance without modifying the model structure. We designed a method that is model-agnostic and can be applied to any graph-based model that consists of user and/or item embedding. More importantly, no trainable parameters are introduced. Its structure is shown in Figure 4(a). We will describe the detailed process in the next paragraph.

First, we need to separately train models with different numbers of layers stacked(three models are trained using the green box frame shown in the Figure 4, with one, two and three layers stacked respectively). Second, each model can get an embedding for a user and an item, and we multiply them to get the score. Three models get three different scores. Since the range of scores obtained by different models may be different, we apply maximum and minimum normalization to limit the scores to the range of 0 to 1. Finally, we fuse the different scores using a weighed summation and define that the value of weight is positively related to the performance of a single model.

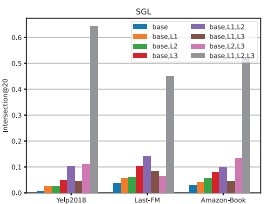

Figure 2: Components of the fused model(SGL).

In order to verify the effectiveness of this method, we use the model trained in section 2.1, and then set the weights of the three models to be 1. The experimental results are shown in Table 2, where %Imp. denotes the relative improvement of the best performing method(bold) over the strongest models(underlined) excluding our method. Our method achieves the best performance. From the point of datasets, our method has obvious improvement effect in Last-FM and Amazon-Book. From the point of the model, SGL has obvious improvement. This illustrate the feasibility and effectiveness of our method.

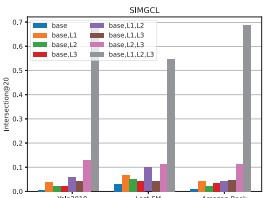

Figure 3: Components of the fused model(SimGCL).

We conduct two experiments to further explore how our method fuses information from different layers and whether new information is

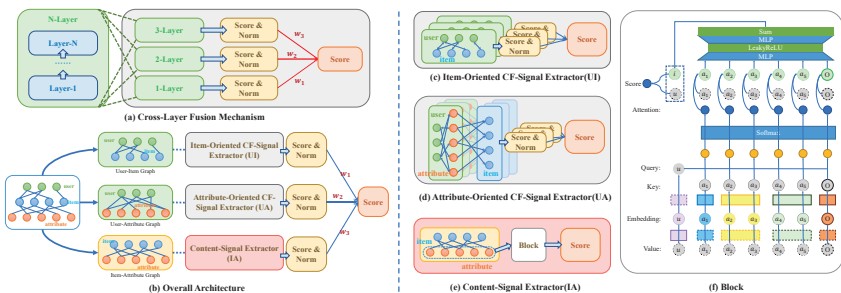

Figure 4: Illustration of the proposed KGSF framework.

generated. The first experiment calculates the `Intersection@20`
between the fused model and model from different layers. The results
are shown in Table 8 in Appendix A(the `Fusion` column indicates that the single model on the left
column is the base, and the `Fusion*` column indicates that the Fusion model is the base). In
the second experiment, statistics are made on which model from different layers the fused model
information comes from. The results are shown in Figure 2 and Figure 3.

From the results, we can get the following conclusions: (1) We can find that the information shared
by three models accounts for the highest proportion after fusion, accounting for about 45% to 69%.
New information accounts for the least after fusion, accounting for about 0.5% to 3.9%. It shows that
the cross-layer fusion mechanism tends to preserve the information shared by three models and will
generate a small amount of new information. (2) In terms of the proportion of retained information,
the information shared by the three models are larger than those shared by the two models (about
4.4% to 14.2%), and the latter is larger than the unique information of a single model (about 2.2%
to 10.0%). This phenomenon is the same as ensemble learning. (3) The retention rate of the method
for a single model information is about 78% to 93%, which means that it can not retain all the
information of the three models. However, the amount of information lost is less than the sum of the
unique information retained and the new information generated, so a better effect is achieved.

## 3 METHODOLOGY

In this section, we introduce our proposed **K**nowledge **G**raph-based **S**eparate and **F**usion model
(KGSF) in detail. First, we decoupling the UIA which used by (Wang et al., 2019c; 2021) into
three graph including UI Graph, UA Graph and IA Graph. Different signals can be found in each
graph. Item-based collaborative filtering signal(i.e., item-user-item co-occurrence) can be found in
UI Graph, attribute-based collaborative filtering signal(i.e., attribute-user-attribute co-occurrence)
can be found in UA Graph and content signal(i.e., items with similar attributes are similar) can be
found in IA Graph. We designed three different signal extractors to mine information. Finally,
we designed a fuser to fuse different signals. It is important to note that this is not an end-to-end
framework. Different signal extractors are trained separately.

### 3.1 ITEM-BASED CF-SIGNAL EXTRACTOR

It is a hot filed to extract collaborative signals in UI Graph for recommendation. Lots of work (Wang
et al., 2019d; He et al., 2020; Wu et al., 2021; Yu et al., 2022; Wang et al., 2020b) has shown that
GNN can effectively extract item-based collaborative signals. Here we adopt the LightGCN (He
et al., 2020), which is a typical and effective model. Its graph convolution operation is defined as:

$$e_{\mathcal{G}_{UI}:u}^{(k+1)} = \sum_{i \in \mathcal{N}_u^{\mathcal{G}_{UI}}} \frac{1}{\sqrt{\left|\mathcal{N}_u^{\mathcal{G}_{UI}}\right|}\sqrt{\left|\mathcal{N}_i^{\mathcal{G}_{UI}}\right|}} e_{\mathcal{G}_{UI}:i}^{(k)}; e_{\mathcal{G}_{UI}:i}^{(k+1)} = \sum_{i \in \mathcal{N}_i^{\mathcal{G}_{UI}}} \frac{1}{\sqrt{\left|\mathcal{N}_i^{\mathcal{G}_{UI}}\right|}\sqrt{\left|\mathcal{N}_u^{\mathcal{G}_{UI}}\right|}} e_{\mathcal{G}_{UI}:u}^{(k)} \quad (1)$$

, where $e_{\mathcal{G}_{UI}:i}^{(k)}, e_{\mathcal{G}_{UI}:u}^{(k)} \in \mathbb{R}^{d_{\mathcal{G}_{UI}}}$ are embedding of item i and user u at layer k, $d_{\mathcal{G}_{UI}}$ is embedding
dimensions, specially $e_{\mathcal{G}_{UI}:i}^{(0)}$ and $e_{\mathcal{G}_{UI}:u}^{(0)}$ are ID embedding (i.e., trainable parameters). We define
$\mathcal{N}_i^{\mathcal{G}_{UI}} = \{u \mid (u, i) \in \boldsymbol{\mathcal{G}_{UI}}\}$ to represent the set of users u who have interacted with item i and

$\mathcal{N}_u^{G_{UI}} = \{i \mid (u,i) \in \mathcal{G_{UI}}\}$ to represent the set of items i that have interacted with user u. Each layer will get embeddings corresponding to items and users. After K layers stacking, LightGCN further combines the embeddings obtained at each layer to form the final representation of a user (an item): $e_i^{\mathcal{G}_{UI}} = \frac{1}{K+1}\sum_{k=0}^{K} e_{\mathcal{G}_{UI}:i}^{(k)}$ , $e_u^{\mathcal{G}_{UI}} = \frac{1}{K+1}\sum_{k=0}^{K} e_{\mathcal{G}_{UI}:u}^{(k)}$.

According to the conclusion in section 2.2, there are differences in the information learned by the models that stack different layers. Therefore, we first train models that stack different layers separately, and then use a cross-layer fusion mechanism for fusion.

## 3.2 ATTRIBUTE-BASED CF-SIGNAL EXTRACTOR

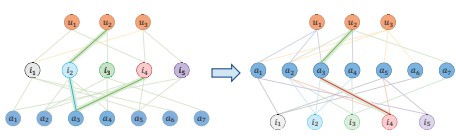

Figure 5: High-Order Connectivity

One of the purpose of introducing KG into recommendation system is to extract attribute-based collaborative filtering signals. Experiments (Wang et al., 2019c; 2021) show that such collaborative signals exist in UIA Graph. As shown in Figure 5, a possible connection path from user $u_2$ to item $i_4$ can be expressed as $u_2 \overset{\text{like}}{\to} i_2 \overset{r_1}{\to} a_3 \overset{r_2}{\to} i_4$. Each node in this path contains the information of other nodes (e.g., the information of node $i_2$ may contain the information of node $a_2$, $a_3$ and $a_4$). But the meaning of introducing this path is to express that user $u_2$ like items containing attribute $a_3$, so this path will introduce noise that is hard to estimate. In order to extract this collaborative signal of users more efficiently, we put the extraction of this collaborative filtering signal on the UA Graph, as shown in Figure 5. At this point, the path can be simplified to $u_2 \overset{\text{like}}{\to} a_3 \overset{r_2}{\to} i_4$. This removes $i_2$ from the original path to avoid introducing unnecessary information, resulting in a purer embedding.

Here, as shown in Figure 4(d) we also apply the LightGCN to extract the collaborative filtering signal from the UA Graph. User and attributes are represented by $e_{\mathcal{G}_{UA}:u}^{(0)}$ and $e_{\mathcal{G}_{UA}:a}^{(0)}$ in layer 0, they are ID embedding (i.e., trainable parameters). According to Formula 1, we can easily define the graph convolution operation in UA Graph as follows:

$$e_{\mathcal{G}_{UA}:u}^{(k+1)} = \sum_{a \in \mathcal{N}_u^{\mathcal{G}_{UA}}} \frac{e_{\mathcal{G}_{UA}:a}^{(k)}}{\sqrt{\left|\mathcal{N}_u^{\mathcal{G}_{UA}}\right|}\sqrt{\left|\mathcal{N}_a^{\mathcal{G}_{UA}}\right|}} ; e_{\mathcal{G}_{UA}:a}^{(k+1)} = \sum_{a \in \mathcal{N}_a^{\mathcal{G}_{UA}}} \frac{e_{\mathcal{G}_{UA}:u}^{(k)}}{\sqrt{\left|\mathcal{N}_a^{\mathcal{G}_{UA}}\right|}\sqrt{\left|\mathcal{N}_u^{\mathcal{G}_{UA}}\right|}}, \quad (2)$$

where $e_{\mathcal{G}_{UA}:a}^{(k)}, e_{\mathcal{G}_{UA}:u}^{(k)} \in \mathbb{R}^{d_{\mathcal{G}_{UA}}}$, $d_{\mathcal{G}_{UA}}$ is embedding dimensions. We define $\mathcal{N}_a^{\mathcal{G}_{UA}} = \{u|(u,a) \in \mathcal{G}_{UA}\}$ to represent the set of users u who have interacted with attribute a and $\mathcal{N}_u^{\mathcal{G}_{UA}} = \{a|(u,a) \in \mathcal{G}_{UA}\}$ to represent the set of attribute a that have interacted with user u. Each layer will get embeddings corresponding to attributes and users. After K layers stacking, LightGCN further combines the embeddings to obtained at each layer to form the final representation of a user or an attribute: $e_a^{\mathcal{G}_{UA}} = \frac{1}{K+1}\sum_{k=0}^{K} e_{\mathcal{G}_{UA}:a}^{(k)}$; $e_u^{\mathcal{G}_{UA}} = \frac{1}{K+1}\sum_{k=0}^{K} e_{\mathcal{G}_{UA}:u}^{(k)}$. Now we have the embedding of user u and attribute a, which contain collaborative signals between users and attributes. In the task description, we talked about getting the user's rating for each item, thus we also need to give the definition of item embedding: $e_i^{\mathcal{G}_{UA}} = \sum_{a \in \mathcal{N}_i^{\mathcal{G}_{UA}}} e_a^{\mathcal{G}_{UA}}$. Likewise, we first train models that stack different layers separately, and then use a cross-layer fusion mechanism for fusion.

## 3.3 CONTENT SIGNAL EXTRACTOR

The second reason for the introduction of knowledge graph is to provide richer information for the embedding of items. The content signal extractor we designed consists of two components. The first is the knowledge graph embedding layer, as shown in Figure 4(e), which uses the knowledge graph embedding method like RotatE (Sun et al., 2018a) to obtain the attribute embedding. The other is the user interest mining layer, as shown in Figure 4(f), its input is the attribute embedding obtained by the first layer, and then using the user as the query vector and the attribute as the key vector, and finally the embedding of the user and item is obtained, denoted as $e_u^{\mathcal{G}_{IA}}$ and $e_{i,u}^{\mathcal{G}_{IA}}$ respectively. For a detailed introduction to these tow layers, please refer to Appendix D.

## 3.4 SIGNAL FUSION MECHANISM

In the previous three subsections, we proposed three different signal extractors. Instead of concatenating between the three different items embedding and users embedding, we train three different signal extractors separately because the three extractors are independent of each other. Here, $S'_{UI}, S'_{UA}, S'_{IA}$ are used to represent the scores of item-based CF-signal extractor, attribute-based CF-signal extractor and content signal extractor, respectively, where $S'_{UI}, S'_{UA}, S'_{IA} \in \mathbb{R}^{|\mathcal{U}| \times |\mathcal{I}|}$. As shown in Figure 4(b), since the three different scores have different ranges, we use max-min normalization to constrain their range between 0 and 1, denoted as $S_{UI}, S_{UA}$ and $S_{IA}$ respectively. The final score is defined as $S = \tau_0 S_{UI} + \tau_1 S_{UA} + \tau_2 S_{IA}$, where $\tau_0, \tau_1, \tau_2 \in (0, 1]$. The value of $\tau_0, \tau_1, \tau_2$ depend on the performance of the three signal extractors. The stronger the performance of the signal extractor, the greater the corresponding weight.

## 3.5 MODEL PREDICTION AND MODEL OPTIMIZATION

In section 3.4, we mentioned that each signal extractor is trained separately, so within each signal extractor, user `u`'s rating for item `i` is defined as the dot product of the corresponding embeddings. The following are the score definitions for each of the three signal extractors.

$$\hat{y}^{\mathcal{G}_{UI}}_{u,i} = e_u^{\mathcal{G}_{UI}^\top} e_i^{\mathcal{G}_{UI}}, \quad \hat{y}^{\mathcal{G}_{UA}}_{u,i} = e_u^{\mathcal{G}_{UA}^\top} e_i^{\mathcal{G}_{UA}}, \quad \hat{y}^{\mathcal{G}_{IA}}_{u,i} = e_u^{\mathcal{G}_{IA}^\top} e_{i,u}^{\mathcal{G}_{IA}}. \tag{3}$$

It should be noted that the scores for item-based cf-signal extractor and attribute-based cf-signal extractor here refer to a model, as shown in the green box in Figure 4(a).

Here, we opt the BPR (Rendle et al., 2012) loss to optimize our model. Since the three signal extractors are trained separately, we give the loss functions of the three signal extractors in Appendix E. It should be noted that the training method in content signal extractor is to train knowledge graph embedding layer first, and then train user interest mining layer alone after the training of knowledge graph embedding layer is completed.

## 4 EXPERIMENTS

We present empirical results to demonstrate the effectiveness of our proposed KGSF framework. The experiments are designed to answer the following research questions: **RQ1**: How does KGSF perform, comparing to the state-of-the-art knowledge-aware recommend models? **RQ2**: How do different components of KGSF (e.g., the attention guiding mechanism, the cross-layer fusion mechanism, the independence and

Table 3: Overall performance comparison.

| | Yelp2018 | | Last-FM | | Amazon-Book | |
|---|---|---|---|---|---|---|
| | Recall | NDCG | Recall | NDCG | Recall | NDCG |
| MF | 6.27% | 4.13% | 7.24% | 6.17% | 13.00% | 6.78% |
| CKE | 6.53% | 4.23% | 7.32% | 6.30% | 13.42% | 6.98% |
| KGNN-LS | 6.71% | 4.22% | 8.80% | 6.42% | 13.62% | 5.60% |
| KGAT | 7.05% | 4.63% | 8.73% | 7.44% | 14.87% | 7.99% |
| CKAN | 6.46% | 4.41% | 8.12% | 6.60% | 14.42% | 6.98% |
| KGIN | 6.98% | 4.51% | 9.78% | 8.48% | 16.87% | 9.15% |
| KGCL | 7.54% | 4.92% | 6.26% | 5.64% | 14.86% | 7.84% |
| HAKG | 7.78% | 5.01% | 10.08% | **9.31%** | - | - |
| **KGSF(ours)** | **8.59%** | **5.60%** | **10.76%** | 9.22% | **17.26%** | **9.46%** |

completeness of three signal extractors, effectiveness of signal fusion mechanism) affect the performance of KGSF? **RQ3**: Can KGSF provide meaningful insights on user intents and give an intuitive impression of explainability? Only some experiments and results are shown here. Please refer to Appendix F for more details.

## 4.1 SETUP

**Datasets, Baselines and Evaluation Metrics.** We use three benchmark datasets: Amazon-Book, Last-FM and Yelp2018, which are extensively evaluated by the SOTA methods (Wang et al., 2019c; 2021; Du et al., 2022; Yuhao et al., 2022) and vary in iterms of domain, size and sparsity. Table 7 presents the overall statistics information of our experimented datasets. We adopt two widely-used evaluation protocols Krichene & Rendle (2022) recall@K and ndcg@K, where K=20 by default. To demonstrate the effectiveness of KGSF, we compare it with the state-of-the-art methods, covering

KG-free method (Rendle et al., 2012), embedding-based method (Zhang et al., 2016), propagation-based methods (Wang et al., 2019a;c; 2021; 2020c) and multiview-based methods (Du et al., 2022; Yuhao et al., 2022). See Appendix F.2 for more details.

## 4.2 PERFORMANCE COMPARISON(RQ1)

We report overall performance evaluation of all methods in Table 3 and performance comparison of KGSF components in Table 4 (`UI` means item-based cf-signal extractor, `UA` means attribute-based cf-signal extractor, `IA` means content signal extractor). It should be noted that dataset Yelp2018 and Amazon-Book apply SimGCL (Yu et al., 2022) in UI components, dataset Last-FM applies SGL (Wu et al., 2021) in UI component. From the results, we summarize the following observations:

Table 4: Performance comparison of KGSF components.

|  | Yelp2018 | | Last-FM | | Amazon-Book | |
|---|---|---|---|---|---|---|
|  | Recall | NDCG | Recall | NDCG | Recall | NDCG |
| UI | 8.39% | 5.50% | 9.31% | 8.10% | 16.03% | 8.68% |
| UA | 3.08% | 1.88% | 9.13% | 7.30% | 7.46% | 4.00% |
| IA | 2.82% | 1.72% | 6.01% | 4.69% | 4.74% | 2.27% |
| UI&UA | **8.56%** | **5.59%** | 10.35% | 8.83% | **16.76%** | **9.05%** |
| UI&IA | 8.44% | 5.52% | **10.36%** | **8.98%** | 16.71% | 9.05% |
| UA&IA | 3.34% | 2.04% | 9.91% | 8.10% | 8.59% | 4.66% |

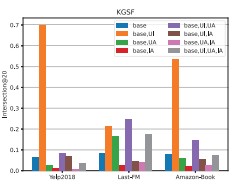

Figure 6: Components of the KGSF.

**(1)** KGSF consistently yields the best performance on the dataset, except for ndcg metric on the Last-FM dataset. In particular, it achieves significant improvement even over the strongest baselines w.r.t. recall@20 by 10.41%, 6.75% and 2.31% in Yelp2018, Last-FM and Amazon-Book, respectively. This demonstrates the rationality and effectiveness of KGSF. We attribute these improvements to fusion mechanism and the decoupling of graphs. **(2)** Jointly analyzing the performance of KGSF across the three datasets, we find that the improvement on Yelp2018 dataset is more significant than that on other dataset. One possible reason is that collaborative signal is more significant to the dataset than the content siganl. This confirms that KSGF has strong adaptability to different types of signals. **(3)** From the perspective of a single component of KGSF, UI component work best, and its performance on Yelp2018 dataset is much higher than the existing SOTA methods that introduce KG. Its performance on Amazon-Book dataset higher than the SOTA methods except KGIN. Therefore, the proportion of item-based collaborative filtering signals in these three dataset is higher than that of the other two signals. If the KG can not be modeled correctly, the introduction of it may weaken the collaborative signal and introduce noise, such as learning on a unified graph structure in KGAT and KGIN, and using a gating mechanism in HAKG to fuse the decoupled embedding, etc.. **(4)** From the fusion of any two components of KGSF, there is no performance degradation and the performance is better than that of a single component. The fusion performance of UI component and UA component is better in most cases. This proves that the introduction of UA graph is effective and it can extract information that can not be extracted by UI graph. The effectiveness of the fusion mechanism is also illustrated.

## 4.3 STUDY OF KGSF(RQ2)

In these section, we present part of our results to explore KGSF. We refer readers to Appendix F for more experiments.

Table 5: Intersection@20 between different components.

|  | Yelp2018 | | | Last-FM | | | Amazon-Book | | |
|---|---|---|---|---|---|---|---|---|---|
|  | UA | IA | UI | UA | IA | UI | UA | IA | UI |
| UA | 1.0000 | 0.2008 | 0.3178 | 1.0000 | 0.2750 | 0.5314 | 1.0000 | 0.2694 | 0.5052 |
| IA | 0.2320 | 1.0000 | 0.3358 | 0.5335 | 1.0000 | 0.5334 | 0.4389 | 1.0000 | 0.4833 |
| UI | 0.1198 | 0.1104 | 1.0000 | 0.4941 | 0.2570 | 1.0000 | 0.2366 | 0.1417 | 1.0000 |

**Independence of three signal extractors.** We judge the independence between each signal extractor, that is, whether the information

extracted by each signal extractor is different. We adopt Intersection@20 to measure independence. Specifically, we calculate the Intersection@20 between the different signal extractors that have been trained. The results are shown in Table 5 (the value of each row is based on the extractor, e.g., 0.2008 in the Yelp2018 indicating that UA component is used as the base, that is $\frac{|UA \cap IA \cap T|}{|UA \cap T|} = 0.2008$). We observe: **(1)** Overall, the value of Intersection@20 between any two component is less than 0.54, indicating that at least 46% of the information extracted between any tow signal extractors is different. **It proves that one of the three signal extractors are independent**. **(2)** Comparing the three datasets, it can be found that the three extractors have the highest independence on Yelp2018 and the lowest independence on the Last-FM. Comparing the performance of each extractors in Table 4, we find that the performance of UA component and IA component is the worst on Yelp2018, and the performance is the best on Last-FM. Therefore, **the independence between individual signal extractors is negatively correlated with the performance between individual signal extractors**. **(3)** In the same dataset, comparing each row, we can find that the independence among the three signal extractors is "IA&UA", "UI&IA" and "UI&UA" in order from strong to weak. At the same time, combined with the results in Table 4, it can be found that the effect of fusion is also in this order from low to high in most cases. This shows that **the stronger the independence, the less the effect will be improved after fusion**.

**Effectiveness of Signal Fusion Mechanism.** We will explore the preservation effect of the signal fusion mechanism and whether new information is generated. The experimental step is to select a benchmark method and then analyze the component proportions of the benchmark method to the three independent signal extractors. The bench-

Table 6: Intersection@20 between KGSF and three components.

|     | Yelp2018 | | Last-FM | | Amazon-Book | |
| --- | --- | --- | --- | --- | --- | --- |
|     | KGSF | KGSF* | KGSF | KGSF* | KGSF | KGSF* |
| UI  | 0.9076 | 0.8884 | 0.7874 | 0.6827 | 0.8547 | 0.8147 |
| UA  | 0.4155 | 0.1544 | 0.7773 | 0.6293 | 0.6949 | 0.3606 |
| IA  | 0.3920 | 0.1264 | 0.6777 | 0.2866 | 0.6443 | 0.1757 |

mark method selected here is KGSF. The experimental results here are shown in Table 6 and Figure 6(c). From the results, we summarize the following observation: **(1)** From the perspective of retention ability, the retention ability of UI component in all datasets is the strongest (about $79\% \sim 91\%$), followed by UA component (about $42\% \sim 78\%$), and the IA component is the worst (about $39\% \sim 68\%$). One possible reason is the setting of $\tau_i$ in the signal fusion mechanism. Within a certain range, the higher the value, the better the retention effect for this component. **(2)** According to Figure 6, we find that about $6\% \sim 9\%$ of the new information is generated in the three datasets. It shows that the signal fusion mechanism will generate new information.

## 5 RELATED WORK

Existing recommendation methods that introduce knowledge graph can be mainly classified into four categories, namely, embedding-based methods, path-based methods, propagation-based methods and multiview-based methods. We give a brief introduction in the Appendix G.

## 6 CONCLUSION AND FUTURE WORK

In this paper, we first propose the Intersection metric to measure the relationship between different models. Experiments verify that the relationship between different collaborative filtering (CF) methods using GNN is shown in Figure 1(b). In addition, the relationship between models using the same CF method but stacking different layers is also the same. We then design a model-agnostic cross-layer fusion mechanism that does not introduce any training parameters, and conduct extensive experiments on three real datasets to demonstrate its effectiveness. Then we analyze the current challenges of introducing KG recommendation methods, and design an extensible KGSF framework to improve the recommendation performance through independent signal extractors and fusion mechanisms, and demonstrate the performance of KGSF on three datasets effectiveness. Experiments show that direct fusion between different signals will interfere with each other, and the current difficulty is how to effectively fuse different features. In the future, we will explore whether the above problems exist in other tasks using GNN, object detection in computer vision, etc., and how to design an efficient, end-to-end feature fusion mechanism.

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

## A PARTIAL FIGURES AND TABLES

Table 7: Statistics of the datasets.

|  | Yelp2018 | Last-FM | Amazon-Book |
|---|---|---|---|
| #Users | 45,919 | 23,566 | 70,679 |
| #Items | 45,538 | 48,123 | 24,915 |
| #Interactions | 1,185,068 | 3,034,796 | 847,733 |
| #Entities | 90,961 | 58,266 | 88,572 |
| #Relations | 42 | 9 | 39 |
| #Triplets | 1,853,704 | 464,567 | 255,746 |

Table 8: Intersection@20 between fusion, SGL and SimGCL.

|  |  | Yelp2018 | | Last-FM | | Amazon-Book | |
|---|---|---|---|---|---|---|---|
|  |  | Fusion | Fusion* | Fusion | Fusion* | Fusion | Fusion* |
| | L-1 | 0.8871 | 0.8138 | 0.8714 | 0.7341 | 0.9032 | 0.7051 |
| SGL | L-2 | 0.9026 | 0.8800 | 0.8633 | 0.7177 | 0.9009 | 0.8097 |
| | L-3 | 0.8396 | 0.8442 | 0.7779 | 0.6992 | 0.8414 | 0.7789 |
| | L-1 | 0.8746 | 0.8192 | 0.8461 | 0.7578 | 0.9106 | 0.8282 |
| SimGCL | L-2 | 0.9065 | 0.8897 | 0.9004 | 0.8107 | 0.9343 | 0.8660 |
| | L-3 | 0.8910 | 0.8779 | 0.8940 | 0.7487 | 0.9151 | 0.8797 |

## B PROBLEM FORMULATION

We first introduce the data structures related to our studied problem, and then formulate our task.

Table 9: Different relations of Intersection

| Intersection@N($M_1$) | Intersection@N($M_2$) | Relation(M1,M2) |
|---|---|---|
| $< 1$ | $= 1$ | Figure1(a) |
| $< 1$ | $< 1$ | Figure1(b) |
| $= 1$ | $= 1$ | Similar |

**User-Item Bipartite Graph (UI).** In this paper, we focus on learning user's preference from implicit feedback including click and purchase. We define user set as $\mathcal{U} = \{u\}$ and item set as $\mathcal{I} = \{i\}$. The user-item bipartite graph is defined as $\mathcal{G}_{UI} = \{(u,i)|u \in \mathcal{U}, i \in \mathcal{I}\}$. If user $u$ has interacted with item $i$, the pair $(u,i)$ will be in $\mathcal{G}_{UI}$.

**Knowledge Graph(KG)/Item-Attribute Graph(IA).** KG stores many real facts, which can express the relationship between entities. They are usually stored in the form of triples. We define $\mathcal{T}$ as a set of triplets, $\mathcal{E}$ as a set of entities, and $\mathcal{R}$ as a set of relations. Let $\mathcal{G}_{IA} = \{(h,r,t)|h,t \in \mathcal{E}, r \in \mathcal{R}\}$ be a collection of triplets, where each $(h,r,t) \in \mathcal{T}$ means that there is a relation $r$ between head entity $h$ and tial entity $t$. For example, a triplet (Wolf Warriors, director, Jason Wu) indicates that the movie Wolf Warriors is directed by Json Wu. Here we assume that all items appear in KG as entities (i.e., $\mathcal{I} \in \mathcal{E}$), which is the common assumption of all existing knowledge-aware recommendation systems. We can connect the items in user-item graph with entities in KG to offer auxiliary semantics to interactions.

**User-Attribute Bipartite Graph(UA).** User-Attribute bipartite graph is actually a combination of User-Item bipartite graph and Item-Attribute graph. Let $\mathcal{A} = \{a|a \in \mathcal{E}, a \notin \mathcal{I}\}$ be a collection of attributes, which is a supplement to items. Let $\mathcal{G}_{UA} = \{(u,a)|u \in \mathcal{U}, a \in \mathcal{A}, (u,i) \in \mathcal{G}_{UI}, (i,r,a) \in \mathcal{G}_{IA}\}$. If user $u$ has been interacted with item $i$ and a relation

$r$ of item $i$ is attribute $a$, the pair $(u, a)$ is in the User-Attribute bipartite graph $\mathcal{G}_{(}UA)$. For example, if user $u$ has watch the movie `Wolf Warriors`, and the director of it is `Json Wu`, so it is said the user $u$ has interacted with `Json Wu`, which can be expressed by `(u,Jason Wu)`.

**User-Item-Attribute(UIA).** This is the same as the **Collaborative Knowledge Graph** definition in KGAT (Wang et al., 2019c).

**Inersection@N.** This is a new metric we propose to measure the difference between two models. Take a user `u` as an example, we consider two models, they will make two ordered lists of items sorted by rating for the user `u`. We remove the ones that appeared in the training set from these two ordered lists and then take out the top N constituent sets, which are recorded as $M_1$ and $M_2$ respectively. We denote the test set as $T$. So let $Intersection@N(M_i) = \frac{|M_1 \cap M_2 \cap T|}{|M_i \cap T|}$, where $|\cdot|$ is the number of elements in the set. The meaning of it is shown in Table 9.

**Task Description.** Given a user-item graph $\mathcal{G}_{UI}$, user-attribute graph $\mathcal{G}_{UA}$ and item-attribute graph $\mathcal{G}_{IA}$, our task is to predict how likely that a user would adopt an item that she or he has never engaged with.

## C    THE RELATIONSHIP BETWEEN SIMGCL AND SGL IN THE SAME LAYER

Table 10: Performance comparison of SGL, SimGCL and their Fusion

|  |  | Yelp2018 | | Last-FM | | Amazon-Book | |
|---|---|---|---|---|---|---|---|
|  |  | SGL* | SimGCL* | SGL* | SimGCL* | SGL* | SimGCL* |
| SGL-1 | SimGCL-1 | 0.6786 | 0.6231 | 0.5214 | 0.5742 | 0.6767 | 0.5834 |
| SGL-2 | SimGCL-2 | 0.7243 | 0.6762 | 0.4965 | 0.5308 | 0.7045 | 0.6768 |
| SGL-3 | SimGCL-3 | 0.7219 | 0.6925 | 0.5080 | 0.6320 | 0.7376 | 0.7045 |

First, we introduce SGL (Wu et al., 2021) and SimGCL (Yu et al., 2022) briefly. SGL proposes three graph argumentation including ND(Node-Drop), ED(Edge-Drop) and RW(Random-Walk) and then applies the InfoNCE (Gutmann & Hyvärinen, 2010) loss function to maximize the similarity of the same node and minimize the similarity of different nodes. SimGCL believes that the graph argumentation of SGL is not necessary. It adopts a simpler method, which is adding noise to nodes to obtain different representations of the same node. Finally InfoNCE is also applied.

Then, we conducted two different experiments under the same conditions (such as dataset, batch size, etc.). It should be noted that graph argumentation method adopted by SGL in the experiment is ED because of its best performance. The first experiment is using recall@20 and ndcg@20 metrics to test the performance of SGL and SimGCL and the results are shown in Table 2. The second experiment is using Intersection@20 metric to measure the similarity of the two methods in the same layer. The results are shown in Table 10 (Take 0.6786 in the dataset yelp2018 as an example, it is represented in the first layer and based on SGL, the similarity between them is 0.6786, that is $\frac{|SGL \cap SimGCL \cap T|}{|SGL \cap T|} = 0.6786$).

From the result, we find that SimGCL performs better on Yelp2018 and Amazon-Book datasets. SGL is better on the Last-FM dataset. In addition, we observe that in Yelp2018 and Amazon-Book datasets, the similarity between the two methods is around 70% in any layer, while the similarity in Last-FM dataset is around 50%. This means that they learned a part of the information that the other did not. From the observation, we can draw the following conclusions: (1) SimGCL is not an improvement over SGL and the relationship between them is intertwined as shown in Figure 1 (a), rather than inclusive or juxtaposed. (2) Both graph argumentation and node perturbation are necessary. This contradicts the conclusions of the SimGCL (Yu et al., 2022).

# D   CONTENT SIGNAL EXTRACTOR

**Knowledge Graph Embedding Layer**.   In the UI Graph, the movies `Wolf Warrior`, `Wolf Warrior 2` and `Mermaid` are independent of each other as three ID embeddings. The introduction of knowledge graph tells us that `Wolf Warrior` and `Wolf Warrior 2` are more similar than `Wolf Warrior` and `Mermaid`, because the former has the same director, genres and actors, while the latter does not. A large number of methods such as TransR (Lin et al., 2015), RotatE (Sun et al., 2018a), etc. have excellent performance in capturing this similarity.

Here we apply the RotatE, which projects entities and relations to the complex plane (i.e., $h, t \in \mathbb{C}^{d_{\mathcal{G}_{IA}}}$, $d_{\mathcal{G}_{IA}}$ is embedding dimension). Specifically, given a triple $(h, r, t) \in \mathcal{G}_{IA}$, the scoring function $d(h, r, t)$ will be as high as possible. If the triple $(h', r, t') \notin \mathcal{G}_{IA}$, the scoring function $d(h', r, t')$ will be as low as possible. The scoring function is defined as $d(h, r, t) = -\|h \circ r - t\|$, where $\circ$ is multiplication in the complex field, $\| \cdot \|$ is L1 norm. RotatE's training method is to consider that the score of valid triplets is higher than broken triplets. Its loss function is defined as follows:

$$\mathcal{L}_{RotatE} = \log \sigma(\gamma + d(h, r, t)) - \sum_{i=1}^{n} p\left(h'_i, r, t'_i\right) \log \sigma\left(-d\left(h'_i, r, t'_i\right) - \gamma\right) \qquad (4)$$

$$p\left(h'_i, r, t'_j \mid \{(h_i, r_i, t_i)\}\right) = \frac{\exp \alpha d\left(h'_j, r, t'_j\right)}{\sum_i \exp \alpha d\left(h'_i, r, t'_i\right)}, \qquad (5)$$

where $\gamma$, $\alpha$ is hyper-parameter, $\sigma(\cdot)$ is sigmoid function, $(h, r, t) \in \mathcal{G}_{IA}$, $(h, r, t) \notin \mathcal{G}_{IA}$. The purpose of introducing $p$ here is to distinguish easy negative samples from difficult negative samples, thereby improving performance.

After training, we will get the embedding representation of the entity $e'_i$, where $e'_i \in \mathcal{E}$. Since $\mathcal{A} \subset \mathcal{E}$, we take out the entity that satisfies $e'_i \in \mathcal{A}$ in the entity set, denoted as $e$. Beacuse $e \in \mathbb{C}^{d_{\mathcal{G}_{IA}}}$, $e$ contains two parts, real part and imaginary part, which we denote as $Re(e)$ and $Im(e)$ respectively. For the convenience of representation, we denote the embedding of entity $e$ as $e_a^{\mathcal{G}_{IA}} = Re(e) \oplus Im(e)$, where $\oplus$ is vector concatenation and $e_a^{\mathcal{G}_{IA}} \in \mathbb{R}^{d_{\mathcal{G}_{IA}} \times 2}$.

**User Interest Mining Layer.** If user `u` likes the item `i`, it must have intention. KGIN (Wang et al., 2021) defines intent as a set of relations, which is coarse-grained. To explore more fine-grained intent, we not only want to know which relationships users are more interested in, but also which attributes users are more interested in. The attention mechanism is the key to realizing this idea. Our idea about attention mechanism is based on Transformer. We take the user as the query vector and the attribute as the key vector to obtain the user's weight for each attribute, and finally aggregate the attributes to obtain the embedding of items. The network architecture of user interest mining layer is shown in Figure 4(f).

The inputs to this layer are $e_a^{\mathcal{G}_{IA}}$ and $e_{other}^{\mathcal{G}_{IA}}$, where $e_a^{\mathcal{G}_{IA}}, e_{other}^{\mathcal{G}_{IA}} \in \mathbb{R}^{d_{\mathcal{G}_{IA}} \times 2}$. The reason for introducing $e_{other}^{\mathcal{G}_{IA}}$ is that not all information can be provided in IA Graph (e.g., the IA Graph only provides the director and actor information of a certain movie, but the user likes the movie because of the genre of the movie, so introducing the `other` attribute can solve this situation).

Considering the same attribute has different relationships with the same item (e.g., the director and actor of `Wolf Warrior` are both `Jason Wu`). In order to model it, we define $\mathcal{R}' = \{id(r) \mid (h, r, t) \in \mathcal{G}_{IA}, h \in \mathcal{J}, t \in \mathcal{A}\} \cup \{id(-r) \mid (h, r, t) \in \mathcal{G}_{IA}, h \in \mathcal{A}, t \in \mathbf{G}_{IA}\}$, where $id(r)$ represents generating a unique, 0-based number for relation `r`. We also define $r(a, i) \in \mathcal{R}'$, which represents the relation id between item `i` and attribute `a`, where $a \in \mathcal{A} \cup \{\text{other}\}$, $i \in \mathcal{J}$. Especially, $r(\text{other}, i) = |\mathcal{R}'|$. It should be noted that if there is no special description, symbol $e_a^{\mathcal{G}_{IA}}$ includes $e_{other}^{\mathcal{G}_{IA}}$

After introducing the introduced symbols, we begin to introduce the user interest mining layer in detail. According to Transformer (Vaswani et al., 2017),TransR (Lin et al., 2015),RGCN (Schlichtkrull et al., 2018), we first give the definition of the value vector and key vector of the attribute:

$$Key_a = W_{key}^{r(a,i)} e_a^{\mathcal{G}_{IA}}, Value_a = W_{value}^{r(a,i)} e_a^{\mathcal{G}_{IA}}, \qquad (6)$$

where $a \in \mathcal{A} \cup \{\text{other}\}$, $i \in \mathcal{J}, W_{key}^{r(a,i)} \in \mathbb{R}^{k \times (d_{\mathcal{G}_{IA}} \times 2)}$, $W_{value}^{r(a,i)} \in \mathbb{R}^{v \times (d_{\mathcal{G}_{IA}} \times 2)}, Key_a \in \mathbb{R}^{k}, Value_a \in \mathbb{R}^{v}$, $k$ and $v$ are the dimension of key vector and value vector of the attribute respec-

tively. Then we define query vector and value vector of user as:

$$Query_u = W_{query}e_u^{\mathcal{G}_{IA}}, Value_u = W_{value}e_u^{\mathcal{G}_{IA}}, \tag{7}$$

where $u \in \mathcal{U}, e_u^{\mathcal{G}_{IA}} \in \mathbb{R}^{d_u}, W_{query} \in \mathbb{R}^{k \times d_u}, W_{value} \in \mathbb{R}^{v \times d_u}, Query_u \in \mathbb{R}^k, Value_u \in \mathbb{R}^v, e_u^{\mathcal{G}_{IA}}$ is trainable parameters.

Here we define $\mathcal{N}_{u,i} = \{a \mid (u,i) \in \mathcal{G}_{UI}, (i,r,a) \in \mathcal{G}_{IA}, u \in \mathcal{U}, i \in \mathcal{I}, a \in \mathcal{A}, r \in \mathcal{R}\}$ to represent the set of attributes of item i that user u have interacted with. Then user u's attention to each attribute of item i is defined as

$$Attention_{u,i} = Softmax\left(\{Query_u^{\top}Key_a \mid a \in \mathcal{N}_{u,i}\}\right) \tag{8}$$

We multiply the attention by the value vector of the corresponding attribute to model the user's attention to the attribute. Different from the previous two extractors, the embedding of the item hear is related to the user, that is, the embedding of the same is different in different users (e.g., given the item i, user $u_1$ and $u_2$ have different degrees of attention to the attributes of the item i, that is, the attention is different, the embedding of the item i will also be different). We define the embedding of item i as:

$$e_{i,u}^{\mathcal{G}_{IA}} = \sum_{a \in \mathcal{N}_{u,i}} ff\left(Attention_{u,i}(a) \cdot Value_a\right), ff(e) = W_{ff}^1\left(LeakyReLU\left(W_{ff}^2 e\right)\right), \tag{9}$$

where $W_{ff}^2 \in \mathbb{R}^{p \times v}, W_{ff}^1 \in \mathbb{R}^{v \times p}, p$ is the dimension of hidden layer, $LeakyReLU(\cdot)$ is activation function. Finally, we define the embedding of user u as $e_u^{\mathcal{G}_{IA}} = Value_u$. Since we added the other attribute, we hope that its attention and other attribute attention are mutually exclusive (i.e., if other's attention are high, the other attribute is low rather than as high as other). To constrain this relationship, we introduce the following loss terms (i.e., guiding attention mechanism):

$$\mathcal{L}_{attention} = \exp\left(\frac{-\left(Attention_{u,i}(other) - \mu\right)^2}{2\sigma^2}\right), \tag{10}$$

where $\mu$ and $\sigma$ are hyper-parameters. This formula is the probability density function of normal distribution. When other's attention value is in the middle, its punishment will be greater. When other's attention value tends to both ends, the punishment will be small.

## E OPTIMIZATION

This section shows the loss function of the three signal extractors. The first two of which consist of BPR and regular terms, and the last one has one more attention constraint than the first two.

$$\begin{aligned}
\mathcal{L}_{UI} &= \sum_{\substack{(u,i) \in \mathcal{G}_{UI} \\ (u,j) \notin \mathcal{G}_{UI}}} -\ln\sigma\left(\widehat{y}_{u,i}^{\mathcal{G}_{UI}} - \widehat{y}_{u,j}^{\mathcal{G}_{UI}}\right) + \lambda^{\mathcal{G}_{UI}}\left\|\Theta^{\mathcal{G}_{UI}}\right\|_2^2 \\
\mathcal{L}_{UA} &= \sum_{\substack{(u,i) \in \mathcal{G}_{UI} \\ (u,j) \notin \mathcal{G}_{UI}}} -\ln\sigma\left(\widehat{y}_{u,i}^{\mathcal{G}_{UA}} - \widehat{y}_{u,j}^{\mathcal{G}_{UA}}\right) + \lambda^{\mathcal{G}_{UA}}\left\|\Theta^{\mathcal{G}_{UA}}\right\|_2^2 \\
\mathcal{L}_{IA} &= \sum_{\substack{(u,i) \in \mathcal{G}_{UI} \\ (u,j) \notin \mathcal{G}_{UI}}} -\ln\sigma\left(\widehat{y}_{u,i}^{\mathcal{G}_{IA}} - \widehat{y}_{u,j}^{\mathcal{G}_{IA}}\right) + \lambda_{attention}^{\mathcal{G}_{IA}}\mathcal{L}_{attention} + \lambda^{\mathcal{G}_{IA}}\left\|\Theta^{\mathcal{G}_{IA}}\right\|_2^2
\end{aligned} \tag{11}$$

where $\Theta^{\mathcal{G}_{UI}} = \left\{e_{\mathcal{G}_{UI}:i}^{(0)}, e_{\mathcal{G}_{UI}:u}^{(0)} \mid i \in \mathcal{I}, u \in \mathcal{U}\right\}, \Theta^{\mathcal{G}_{UA}} = \left\{e_{\mathcal{G}_{UA}:a}^{(0)}, e_{\mathcal{G}_{UA}:u}^{(0)} \mid a \in \mathcal{A}, u \in \mathcal{U}\right\}$ and $\Theta^{\mathcal{G}_{IA}} = \{e_{other}^{\mathcal{G}_{IA}}, e_a^{\mathcal{G}_{IA}}, e_u^{\mathcal{G}_{IA}}, W_{key}^{r(a,i)}, W_{value}^{r(a,i)}, Query_u, Value_u, W_{ff}^1, W_{ff}^2 \mid a \in \mathcal{A}, u \in \mathcal{U}, r(a,i) \in \mathcal{R}'\}$ are the set of model parameters. $\lambda^{\mathcal{G}_{UI}}, \lambda^{\mathcal{G}_{UA}}, \lambda^{\mathcal{G}_{IA}}$ and $\lambda_{attention}^{\mathcal{G}_{IA}}$ are hyper-parameters to control the $L_2$ regularization term and attention loss, respectively.

# F ADDITIONAL EXPERIMENTAL SETUPS AND RESULTS

## F.1 BASELINES

**MF** (Rendle et al., 2012) is a typical matrix factorization method and doesn't use KG information. It uses ID embeddings of users and items to make the prediction in implementation.

**CKE** (Zhang et al., 2016) is a representative embedding method. It utilizes the TransR (Lin et al., 2015) to encode entities in the KG, which are then used as input to MF framework.

**KGNN-LS** (Wang et al., 2019a) is a propagation-based model, which converts KG into user-specific graphs, and then considers user preference on KG relations and label smoothness in the information aggregation phase, so as to generate the different representation of the same item under different users.

**KGAT** (Wang et al., 2019c) is a propagation-based recommend model. It applies a unified relation-aware attentive aggregation mechanism in UIA to generate user and item representations.

**CKAN** (Wang et al., 2020c) is based on KGNN-LS, which utilizes different aggregation schemes on the user-item graph and KG respectively, to mine knowledge association and collaborative signals.

**KGIN** (Wang et al., 2021) is a state-of-the-art propagation-based method, which models user interaction behaviors with latent intents, and proposes a relation-aware information aggregation scheme to capture long-range connectivity in KG.

**HAKG** (Du et al., 2022) is also a state-of-the-art multiview-based method which embed users, relations and items in hyperbolic space and use a hyperbolic aggregation scheme. It learns from UI and IA graph to generate collaborative signals and knowledge associations, and applies gating mechanism to fuse them.

**KGCL** (Yuhao et al., 2022) is a general contrastive learning framework using knowledge graph augmentation schema. Besides, it leverages additional supervision signals to guide a cross-view contrastive learning paradigm.

## F.2 PARAMETER SETTINGS

For a fair comparison, we fix the size of ID embeddings as 64 (except that the embedding of the UA component on the Amazon-Book dataset is 128), the optimizer as Adam (Kingma & Ba, 2014), and the batch size as 4096 for all methods. The Xavier (Glorot & Bengio, 2010) initializer is used to initialize the model parameters. We consider learning rate $lr \in \{10^{-4}, 10^{-3}, 10^{-2}\}$, $\tau_0, \tau_1, \tau_2 \in \{0.1, 0.2, ..., 1.0\}$, $\lambda^{\mathcal{G}_{UI}}, \lambda^{\mathcal{G}_{UA}}, \lambda^{\mathcal{G}_{IA}} \in \{10^{-3}, 10^{-4}, 10^{-5}\}$, $\mu = 0.5$, $\sigma = 0.15$, $\lambda^{\mathcal{G}_{IA}}_{\text{attention}} = 10^{-5}$, $p = 128$.

The parameters for all baseline methods are carefully tuned to achieve optimal performance. Specifically, for KGAT (Wang et al., 2019c), we set the depth to three with the hidden size $\{64, 32, 16\}$, and use the pre-trained ID embeddings of MF (Rendle et al., 2012) as initialization; for CKAN (Wang et al., 2020c), KGNN-LS (Wang et al., 2019a), we set the size of neighborhood to 16; for KGIN (Wang et al., 2021), we fix the number of intents to 4; Moreover, early stopping strategy is performed for all methods, i.e., premature stopping if recall@20 on the test set does not increase for 10 successive epochs.

## F.3 ADDITIONAL STUDY OF KGSF(RQ2)

Since graph decoupling, independent training and signal fusion mechanism are the core of KGSF, we conduct extensive experiments to explore their effectiveness. Specifically, we first analyze individual components, including the attention guiding mechanism of the IA component and the cross-layer fusion mechanism of the UA component. We then delve into the independence and completeness of each signal extractor.

**Impact of Attention Guiding Mechanism in IA Graph**. Here we verify the effectiveness of the attention guiding mechanism, thus we design two variants of the IA component. A variant discards the attention guiding mechanism, denoted as "IA w/o G", where $\lambda^{\mathcal{G}_{IA}}_{attention} = 0$. Another variant retains the attention guiding mechanism, denoted as "IA", where $\lambda^{\mathcal{G}_{IA}}_{attention} = 1$. The experimental

Table 11: Impact of Attention Guiding Mechanism.

|  | Yelp2018 | | Last-FM | | Amazon-Book | |
|---|---|---|---|---|---|---|
|  | Recall | NDCG | Recall | NDCG | Recall | NDCG |
| IA w/o G | 2.69% | 1.66% | 5.81% | 4.53% | 4.70% | 2.23% |
| IA | 2.82% | 1.72% | 6.01% | 4.69% | 4.74% | 2.27% |

results are shown in Table 8. Degraded performance can be observed in all datasets, indicating the necessity of the attention guiding mechanism. Specifically, if the attention guiding mechanism is not introduced, the model will preferentially optimize the Other embedding, resulting in the Other's attention close to other attribute's attention, which is not helpful for the interpretability of the model. Introducing the attention guiding mechanism can make the model pay more attention to the information of other attribute.

Table 12: Impact of the number layers L and Fusion.

|  | Yelp2018 | | Last-FM | | Amazon-Book | |
|---|---|---|---|---|---|---|
|  | Recall | NDCG | Recall | NDCG | Recall | NDCG |
| UA-1 | 2.88% | 1.77% | 8.19% | 6.42% | 6.97% | 3.65% |
| UA-2 | 2.98% | 1.82% | 8.47% | 6.75% | 6.98% | 3.79% |
| UA-3 | 2.64% | 1.60% | 7.86% | 6.21% | 5.83% | 3.00% |
| Fusion | **3.08%** | **1.88%** | **9.13%** | **7.30%** | **7.46%** | **4.00%** |

**Impact of model depth and Cross-Layer Fusion Mechanism in UI component.** Here, we search for L in range $\{1, 2, 3\}$ and then use cross-layer fusion mechanism to fuse them. We use "UA-i" to represent UA component of the stacked i layer and use "Fusion" to represent the model with the cross-layer fusion mechanism. The experimental results are shown in Table 12. Our observations are as follows:

- In the three datasets, the performance of "UA-2" is greater than that of "UA-1" and the performance of "UA-1" is greater than that of "UA-3". One possible reason for this is that the UA component can simplify high-order connectivity proposed by KGAT. In KGAT, it is necessary to stack 3 to 4 layers to achieve better results, while UA component can achieve better results after stacking 2 layers.

- According to the conclusion in Section 2.1, the information extracted by UA component at different layers is different. We applied the cross-layer fusion mechanism and found that the performance was better than that of a single model. The improvement was obvious on the Last-FM and Amazon-Book datasets. This further verifies the existence of partial information independence in different layers and the effectiveness of the cross-layer fusion mechanism.

Table 13: Intersection@20 between KGIN and three components.

|  | Yelp2018 | | Last-FM | | Amazon-Book | |
|---|---|---|---|---|---|---|
|  | KGIN | KGIN* | KGIN | KGIN* | KGIN | KGIN* |
| UI | 0.4002 | 0.5588 | 0.5725 | 0.5900 | 0.5501 | 0.6144 |
| UA | 0.3852 | 0.2039 | 0.5212 | 0.4953 | 0.5318 | 0.2725 |
| IA | 0.3680 | 0.1699 | 0.5558 | 0.2734 | 0.5042 | 0.1617 |

**Completeness of Three Signal Extractors**. We judge the completeness of each signal extractor, that is, whether these three signal extractors can extract all information after the introduction of KG. Our experimental setup is to select a benchmark method and then analyze the component proportions of the benchmark method to the three independent signal extractors.

The benchmark method selected here is the SOTA method KGIN (Wang et al., 2021). The experimental results are shown in Figure 6(b) and Table 13. The Inverse in the Table 11 means KGIN[9]

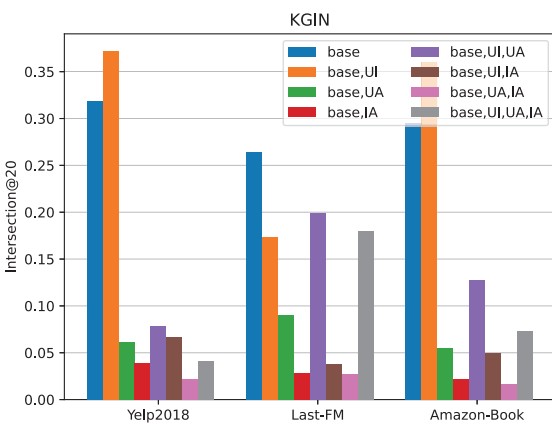

Figure 7: Components of the KGIN.

is the base, and the KGIN means the column element is the base. Taking 0.5588 as example, 0.5588 in Yelp2018 dataset means $\frac{|UI \cap KGIN \cap T|}{|KGIN \cap T|} = 0.5588$. We can observe as follows:

- According to Figure 7, we find that about 30% of the information is not extracted by the three signal extractors. The UI component and UA component apply the cross-layer fusion mechanism. According to the conclusion in Section 2.2, the cross-layer fusion mechanism will lose some information. Therefore, we don't use the cross-layer fusion mechanism, so the original signal base extractors are changed from three ("UI", "UA", "IA") to seven ("UI-1", "UI-2", "UI-3", "UA-1", "UA-2", "UA-3", "IA"). The experimental results show that about 25% of the information does not exist in the three signal extractors. This shows that the information in the three signal extractors cannot extract all information, only convers 75% of the information. Therefore the three signal extractors are not completeness. A possible reason is that the relationship is not preserved in the three signal extractors.

- According to Table 13, we can find that the components of KGIN (Wang et al., 2021) are UI, UA and IA in descending order, among which UI components accounts for about 60% of the three datasets. From the perspective of UI component, its retention rate is somewhere between 40% to 50%. This shows that the end-to-end method of KGIN cannot completely retain the collaborative filtering signal of user-item. The reason why the performance of KGIN] is better than collaborative filtering is that the discarded information is smaller than the new information learned.

### F.4   EXPLAINABILITY OF KGSF (RQ3)

Benefiting from the separate modeling and fusion mechanism of KGSF, we can get more fine-grained explanations than KGIN. We randomly selected user $u_{335}$ and a related item $i_{4079}$ (from the test, unseen in the training phase), which is a book called Never Go Back. The interpretability obtained after visualizing it is shown in Figure 8. We have the following findings:

- From Figure 8(a), we can find that the user $u_{335}$ likes this book not because of the item-based collaborative signal, but because of the attribute-based collaborative signal or content signal.

- Figure 8(b) and Figure 8(c) are the user's overall preference and preference for a certain attribute of the item $i_{4079}$ generated by the UA component, respectively. Figure 8(b) shows that after the dot product between the user and all attributes, we select the 10 attributes with largest values as the most favorite 10 attributes of users. We can find that all 10 attributes are writers, and the writing styles of writers tends to styles like thriller, crime, mystery, etc., which we can think of as user preferences. Figure 8(c) shows the first three attributes obtained by normalizing the attributes of the item $i_{4079}$ and the user through dot product.

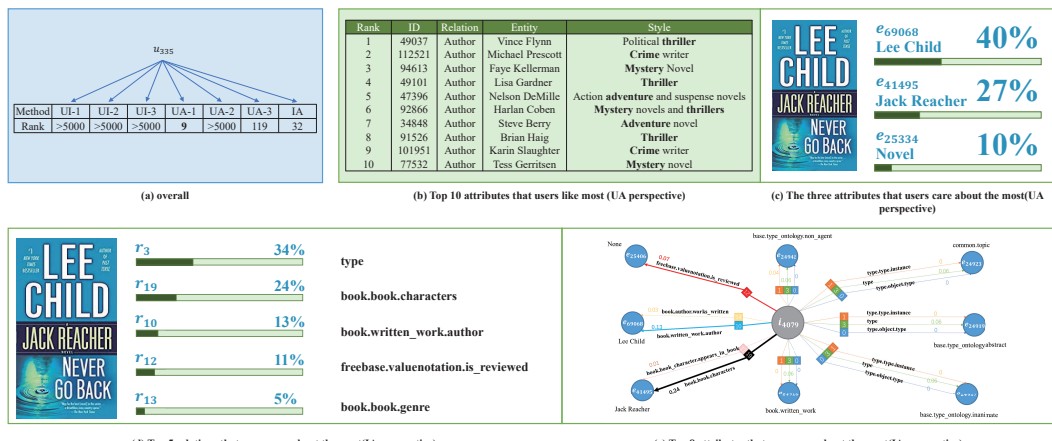

Figure 8: Explanations of user intents and real cases in Amazon-Book.

It can be found that 40% of the reasons why user $u_{335}$ likes item $i_{4079}$ are the author Lee Child, 27% of the reasons are the protagonist "Jack Reacher" in the book, and 10% of the reasons are because the book is a novel.

- Figure 8(d) and Figure 8(e) are the user's attention to the relationship of this book and the attribute attention of this book generated by IA component, respectively. It can be seen that the reason why user $u_{335}$ likes the book is because of the type of the book, the character in the book and the author of the book, which is similar to the conclusion in Figure 8(c). The two separately trained models get the same result, it shows that this interpretability is credible. Figure 8(e) gives detailed attention. For example, user prefers that the character in the book is "Jack Reacher" in the book (attention for path Nerver Go Back $\xrightarrow{\text{characters}}$ Jack Reacher is 0.24), instead of "Jack Reacher" appears in the book (attention for path Jack Reacher $\xrightarrow{\text{appears}}$ Nerver Go Back is only 0.01). This is very intuitive.

# G  RELATED WORK

**Embedding-based methods**  (Zhang et al., 2016; Cao et al., 2019a; Ai et al., 2018; Cao et al., 2019b; Huang et al., 2018; Wang et al., 2020a; 2018b) first use knowledge graph embedding techniques (e.g., TransR (Lin et al., 2015), RotatE (Sun et al., 2018a)) to obtain entity and relation embeddings, and then input these embeddings into subsequent recommendation networks. For example, CKE (Zhang et al., 2016) uses the TransR (Lin et al., 2015) to learn the structural information of entities from the knowledge graph, and then inputs the learned embeddings into matrix factorization (MF (Rendle et al., 2012)). KTUP (Cao et al., 2019a) applies TransH (Wang et al., 2016) on both the user-item bipartite graph and the knowledge graph to jointly learn user preferences and complete recommendations. Although these methods can capture the similarity between entities brought by the KG, they ignore the information brought by higher-order connectivity.

**Path-based methods**  (Catherine & Cohen, 2016; Hu et al., 2018; Jin et al., 2020; Ma et al., 2019; Wang et al., 2019e; Sun et al., 2018b) find higher-order connectivity for recommendation by finding semantic paths in the KG and then connecting items and users. These paths can be input to RNN network (Wang et al., 2019e; Sun et al., 2018b) or employ an attention mechanism (Hu et al., 2018) to extract user preferences. For example, KPRN (Wang et al., 2019e) infer the potential high-order connectivity of a user-item interaction by mining the sequential dependence within a knowledge-aware path. But defining the correct meta-path requires domain knowledge, which is labor-intensive and time-consuming for KG with multiple relationships and various types of entities. At the same time, recommendation systems applied in different fields cannot be transferred to each other, and the generalization ability is poor.

**Propagation-based** (Wang et al., 2021; 2019c; Tu et al., 2021; Wang et al., 2018a; 2019a;b; 2020c) methods rely on the information aggregation ability of GNN and the stacking ability of layers to capture high-order connection in an end-to-end manner automatically. For example, KGAT (Wang et al., 2019c) introduces an attention mechanism on unified UKG graph for learning. Based on KGAT (Wang et al., 2019c), KGIN (Wang et al., 2021) transfers relational information and introduces intent nodes between users and items to achieve better interpretability and performance. Although this unified graph structure captures collaborative filtering signals, high-order connectivity and knowledge information. It is experimentally proved that only part of this information can be captured. The reason why it is not fully captured is that these information are mixed together when propagating on the graph, which introduces an incalculable noises.

**Multiview-based methods** (Zou et al., 2022; Yuhao et al., 2022; Du et al., 2022) choose to construct multiple views to learn information from different perspectives, and then learn embeddings about items and users by designing a fusion mechanism. For example, HAKG (Du et al., 2022) learns from user-item graph and knowledge graph to generate collaborative signals and knowledge associations, and applies gating mechanism to fuse them. Although this method achieves better performance than propagation-based methods, it still cannot lean the full information of these views because different information is still represented by one embedding.

