# OpenReview forum: "HAS IT REALLY IMPROVED? KNOWLEDGE GRAPH BASED SEPARATION AND FUSION FOR RECOMMENDATION"
_ICLR.cc/2023/Conference — Submitted to ICLR 2023_

### Official Review · Reviewer_8G5y · 2022-10-22

**Confidence:** 5
**Correctness:** 3
**Technical Novelty And Significance:** 2
**Empirical Novelty And Significance:** 2
**Recommendation:** 3

**Clarity, Quality, Novelty And Reproducibility:**

Overall, this paper needs to be re-organized to make it mainly focus on knowledge graph-based recommendation methods, instead of traditional GNN-based recommendation methods. The technical novelty of the proposed KGSF framework is limited.

**Strength And Weaknesses:**

Strengths:
1.	The authors propose a new metric, i.e., Intersection@N, to measure the differences between two models. This is an interesting idea.

2.	The authors propose a cross-layer fusion mechanism to improve the performance of GNN models.

3.	The authors propose a knowledge graph-based recommendation framework, i.e., KGSF, to exploit item knowledge graph for recommendation.

Weaknesses:
1.	The organization of this work is a little confusing. It seems this paper should focus on knowledge graph-based recommendation methods. However, the authors spent lots of efforts analyzing traditional GNN based methods, for example the analysis in Table 1, Table 2, Figure 2, and Figure 3.  The authors need to reorganize the structure of this work.

2.	The proposed recommendation framework is just an integration of existing methods. The technical novelty of this work is very limited.

3.	The proposed KGSF recommendation framework considers three graphs for recommendation, i.e., UI graph, IA graph, and UA graph. Besides the user-item interaction information, KGSF can only exploit the item attribute information in knowledge graph. Thus, it can not exploit the complex structure information and other entity information.


**Summary Of The Paper:**

This paper studies the knowledge graph-based recommendation problem. The authors firstly study the relationship between different SOTA methods. The authors also develop a model-agnostic cross-layer fusion mechanism to improve the performance of GNN. To demonstrate the effectiveness of the proposed method, the authors have performed experiments on three public datasets.

**Summary Of The Review:**

This paper firstly proposes a metric to evaluate the relationship between two models. Then, the authors also propose a cross-layer fusion mechanism to improve the performance of GNN models. Moreover, the authors also propose a knowledge graph based recommendation framework KGSF. However, the connections between these three parts are not very clear. Simply putting these three parts together makes the focus of this paper is not very clear. Moreover, the technical contribution of the proposed KGSF model is limited. The following are some other detailed comments.

1. The font size in Figure 1 is too small. It is better to use a larger font size.

2. Intersection@N is one of the main contributions of this work. The authors need to introduce the definition of Intersection@N and discuss its reasonability in the main content of this work.

3. In this work, the authors study knowledge graph-based recommendation. Knowledge graph is generally a complex graph which does not only include item attributes. However, in this work, the authors only consider item attributes in the knowledge graph. This assumption seems not reasonable enough.

4. It seems that this work should mainly focus on knowledge graph-based recommendation methods. However, the authors have spent many efforts studying traditional GNN-based recommendation methods, e.g., SGL and SimGCL. This makes this work a little confusing.

5. In Table 3, the authors need to include the performance achieved by LightGCN on all the experimental datasets.

6. In Section 4.1, the authors also need to introduce the detailed strategies used to split the train, validation, and test datasets.

---

### Official Review · Reviewer_wbd8 · 2022-10-23

**Confidence:** 4
**Correctness:** 2
**Technical Novelty And Significance:** 2
**Empirical Novelty And Significance:** 2
**Recommendation:** 3

**Clarity, Quality, Novelty And Reproducibility:**

Clarity: below average.

Quality: below average.

Novelty: marginally good.

Reproducibility: should be reasonably possible to reproduce the work mentioned in the paper.

The writing of the paper needs a major improvement to increase clarity. I list a few concrete examples below (but not all) plus some minor comment on formatting and typos.

==== missing clarity

- "items" and "commodities" are both used. I would recommend we only use "items" to avoid confusion.
- M1 learns new things on the basis of retaining the information of M2. In your Figure 1 (a), M1 is not a superset of M2, it only retains the information of M2 when the test set is considered as the reference.
- It's confusing to say that "models that stack higher layers cannot fully 'include' models that stack lower layers.". I did not seem to understand what you mean by "include".
- There are multiple mentioning of "findings" or "observations" in the introduction. I recommend that you summarize your findings or observations clearly in the beginning and then elaborate them.
- Two categories of improvement. I think in reality, the first category is almost impossible. In the experiment result that the author shows, I did not see that any "better model" could retain 100% of information from a base model.
* Table 2. in the table it uses "Imp%", in the text, it says "%Imp", we shall be consistent on the notation.
* section 2.2. "we multiply them to get the score". Please be precise in your language to describe how the score is obtained. Do you mean element-wise multiplication between user embedding and item embedding or a dot-product between the two embeddings?
* section 3.1. "hot filed" -> "hot field" ?
* section 3.4. "The value of τ0, τ1, τ2 depend on the performance of the three signal extractors". Please clarify how to determine the performance of each individual extractor. Is that through a validation dataset?
* section 4.3. "max-min normalization". I think this is usually called "min-max normalization" instead (see https://en.wikipedia.org/wiki/Feature_scaling#Rescaling_(min-max_normalization).

=== Some of claims are not well-supported.
* "To objectively measure these two cases, we design a new metric, Intersection@n, to measure the differences between two models". Would you give a few more explanations why you think that this new measure is objective? Actually most measures could be biased in some sense though they could be useful. I won't claim this is an objective measure without giving the assumptions it is based on.
* section 2.2. "can be applied to any graph-based model". Please give supportive evidence before making such claims.
* section 4.3. I cannot see why your observations can lead to the conclusion that "one of the three signal extractors are independent". You might want to clarify what you mean by "independent" if it is different from the usual statistical meaning of it.




**Strength And Weaknesses:**

Strength:
* The proposed metric `intersection@n` is somewhat novel to me. In practice, it could be useful to measure the model improvement when comparing a candidate model with a baseline model if $N$ is evaluated at various levels (e.g, 5, 10, 20, 100). This evaluation is especially useful when we need a validation measure before deploying new model to replace the previous model because the recommendation model often needs an update (either continuously or after a fixed interval of time).

Weakness:

* The technical contribution of this paper is minor, due to the fact that the fusion mechanism introduced in this paper does not seem novel to me.
* The writing of the paper is not clear in many places. I will list some examples below.
* The author uses a large portion of the paper to describe their analysis using the proposed "intersection@n", which in my opinion, does not seem necessary. Because this part is less technical and not relevant to the modeling contribution they described thereafter, which shall probably be placed in the appendix instead, and more technical discussions could be used for the main text.
* Most figures (e.g., figure 1, 2, 3, 4) have used very small fonts with illegible texts. I have to zoom to 3x to see clearly the text in the figures. This is not friendly for a hard print on normal A4 paper.


**Summary Of The Paper:**

This paper studies recommendation models based on knowledge graph. The author proposes a metric to measure the candidate model prediction performance compared to a base model. The findings throughout evaluating the metric against a few prior models are used to motivate their own work, in which they design a model-agnostic fusion mechanism to improve GNN performance (with multiple layers). They also propose the idea to use three independent signals extractors and later fuse them together to perform recommendations. Their empirical experiments show improvements over previous KG-based methods.

**Summary Of The Review:**

The paper has limited technical contribution and could use more improvement in the writing. It does not meet the bar of acceptance at its current form.

---

### Official Review · Reviewer_3bRV · 2022-10-25

**Confidence:** 4
**Clarity, Quality, Novelty And Reproducibility:** 1. The novelty of this work is limite…
**Correctness:** 3
**Technical Novelty And Significance:** 1
**Empirical Novelty And Significance:** 3
**Recommendation:** 3

**Strength And Weaknesses:**

Strength:

1. The idea of separating and fusing KG and recommender systems is interesting;

2. The paper is basically well-written and well-organized.

Weaknesses:

1. The novelty of this paper is limited;

2. Some of the technical details are not clearly presented;

3. The empirical improvement over baseline methods are not significant.

See below for details of weaknesses.


**Summary Of The Paper:**

This paper first proposes the Intersection metric to measure the relationship between different KG-based recommender system models. The authors then design a model-agnostic cross-layer fusion mechanism, and conduct experiments on three real datasets to demonstrate its effectiveness. At last, the authors design a KGSF framework to improve the recommendation performance through independent signal extractors and fusion mechanisms, which works on user-attribute, item-attribute, and user-item graphs, separately.

**Summary Of The Review:**

Overall, the contribution of this paper is limited, and the experimental result does not show significant improvement over baseline methods. In my opinion, the paper does not reach the acceptance threshold of ICLR, and I vote for reject.

---

### Decision · Program_Chairs · 2023-01-20

**Decision:**

Reject

**Justification For Why Not Higher Score:**

Technical contribution is minor

**Justification For Why Not Lower Score:**

N/A

**Metareview: Summary, Strengths And Weaknesses:**

Authors investigates recommendation models based on knowledge graph where they have proposed a metric intersection@n to measure the candidate model prediction performance and designed a model-agnostic fusion mechanism to improve GNN performance. They have performed their empirical experiments to demonstrate they outperform existing KG-based methods.

The main limitation of this paper is its technical contribution is not significant. In addition, the paper is not well written and reviewers cannot understand very well.


**Summary Of Ac-Reviewer Meeting:**

No need, every reviewers do not like this paper